# MEC-Enabled Fine-Grained Task Offloading for UAV Networks in Urban Environments

**Sicong Yu, Huiji Zheng and Caihong Ma \***

Information Engineering College, Engineering University of PAP, Xi'an 710018, China
* Correspondence: mach7892022@163.com; Tel.: +86-151-2936-0116

**Abstract:** In recent years, with the continuous development of information technology, the amount of data generated and hosted by cloud service platforms in urban environments is unprecedented. Mobile edge computing (MEC) is combined with UAV networks to better realize the ability to provide nearby services to a large number of terminal devices in cities. Unmanned aerial vehicles (UAVs) are highly maneuverable and inexpensive and are good carriers for carrying MEC platforms. In UAV edge networks, we usually face the problem of fine-grained task offloading based on relevant features of urban environments. We need to address high energy consumption and task processing delays to help achieve urban sustainability goals. Therefore, we combine the software definition network (SDN) technology and, on this basis, we propose two task offloading strategies based on an improved EFO intelligent algorithm for different user scales. At the same time, we run the proposed offloading system in the UAV sensor. The experiment shows that, compared with the traditional strategy, the unloading efficiency of the proposed method can be improved by about 10%.

**Keywords:** UAV; MEC; sustainable urban living; SDN; EFO algorithm





## 1. Introduction

### 1.1. Motivation

In recent years, with the rise of information technology, the explosive growth of Internet of Things (IoT) devices and massive data have led to the rapid development of new network systems such as s smart cities [1], Internet of Vehicles (IoV) [2], and wireless sensor networks [3,4]. Especially in urban environments, with the continuous development of information technology, the amount of data generated and hosted on cloud service platforms in urban environments is unprecedented. According to the forecast of HIS Markit, by 2025, the number of IoT devices will exceed 75 billion, which will bring unprecedented energy consumption pressure and delay problems to the traditional cloud service model [5]. Based on this, MEC began to be applied to the cloud service model in the urban environment. This is a new distributed computing model proposed by the European Telecommunications Standardization Institute (ETSI) in 2014 to serve mobile users at the edge of the network. Because it can meet the real-time, intelligence, and security requirements of network edge users, once it was proposed, it quickly became the focus of academic and business circles. The core idea of MEC is to decentralize the computing resources originally on the central cloud to the edge of the network, so that end users can obtain computing services near the edge. The goal is to optimize system latency [6], energy consumption [7], and the total cost of energy consumption and latency [8] in urban environments. However, all these studies assume that wired or dedicated wireless connections have sufficient bandwidth in distributed edge resources deployed in a fixed fashion. In particular, existing MEC techniques are not suitable for situations where the number of mobile users increases explosively or the network facilities are sparsely distributed [9]. Based on this, UAV technology can be used as a solution to the problem of terrestrial wireless network connectivity

In green smart cities, drones are widely used. They usually fly around the city and use their own sensors to perform specific tasks, such as target tracking [10], network coverage [11], relay communication [12], trajectory planning [13], etc. Generally speaking, in the traditional UAV edge network, the UAV platform has difficulty carrying high-load, high-density computing-intensive tasks due to its limited computing power. As a result, a single drone is often used as a data collection terminal, which collects data, uploads it to a remote central cloud, and then uses its nearly unlimited computing resources and computing power to process the data and send it back. This approach makes up for the weak computing power and scarce computing resources of UAVs. However, based on the high energy consumption and high latency problems faced by traditional cloud services in the era of big data explosion, UAV platforms urgently need stable short-distance services to ensure lower transmission latency.

Therefore, the combination of UAV and MEC technology has become a good solution for achieving energy saving and low-latency goals for green smart cities. UAVs have become an ideal platform for carrying MEC servers due to their excellent mobile performance, flexible deployment, small size, and concealed operation.

After the introduction of MEC, we still need to consider the following issues according to the urban environment:

- Dynamic deployment of drone platforms: The city is in a dynamic environment, and mobile intelligent devices will access and exit the network at any time. Therefore, the cloud service model is required to have strong dynamic perception and the ability to adapt to the external environment, which requires the UAV platform to improve the ability to dynamically deploy according to user conditions and puts forward higher requirements for network flexibility.
- Uninstallation decision-making problem: With the continuous enhancement of the computing and storage capabilities of mobile smart devices, some computing tasks can be carried out in real time on mobile smart devices in the urban environment, which promotes the evolution of computing models from centralized to distributed [14]. However, mobile smart devices are unable to complete computing-intensive and time-sensitive tasks alone because of the rather limited resources. Therefore, task offloading and collaboration computing are needed to offload part or all of the computing tasks to the MEC platform on the UAV. The edge node can not only directly process the computing task and then return the execution result but also pre-process the task before delivering it to the cloud center for processing in the case of good network conditions and heavy task load. Afterwards, when performing task offloading, we need to compare the time latency of offloading tasks with different processors under certain cost constraints, so that the question of where the task is to be offloaded (that is, to the local or the edge server using the UAV platform as the carrier) is considered.
- Fine-grained problem of task offloading: In urban environments, most tasks that need to be processed can generally be decomposed into sub-tasks of different sizes, with certain context dependencies or parallelism. For example, in route navigation, it can be simply divided into "confirm the target–confirm the traffic–confirm the route". The offloading process for such tasks is called fine-grained task offloading.

To solve the above problems, this paper explores the management and control of the entire network based on the UAV edge network, combined with SDN technology, and improves the reliability and attempts to flexibility of the network. At the same time, this paper models the problem of computing task offloading between end users and MEC servers and proposes a task offloading strategy based on an improved EFO algorithm for two different user-scale scenarios.

### 1.2. Related Work

Faced with the ever-increasing amount of mobile data, emerging computing-intensive user applications place higher demands on the security [15], flexibility and resilience of cloud/edge architectures. Based on this, the UAV network combined with MEC as a

new type of network architecture has received extensive attention from the academic community. Cheng et al. in [16] proposed a new type of air–ground integrated mobile edge network (AGMEN), in which UAV, as the MEC node of the edge network, can realize the flexible deployment and scheduling of resources and assist the remote cloud to fulfill functions such as caching, computing, and communication. A UAV–edge–cloud hybrid computing model proposed by Chen et al. in [17] makes full use of the characteristics of UAV clusters, edge layers, and remote clouds, and the correlation of the three enables UAV clusters to significantly reduce energy consumption, latency, and operating costs, as well when processing high-density computing tasks. Cao et al., in [18], discussed an MEC service model consisting of UAVs and cellular ground base stations, in which a single UAV offloads computing tasks to five ground base stations along the way from its initial location to its final location scene. In the case of meeting the maximum speed limit of the UAV and the limited computing capacity of the ground base station, the UAV path and unloading decision-making scheme are jointly optimized by using the alternating optimization and successive convex approximation technology, so as to minimize the task completion time of the UAV. Zhang et al., in [19], constructed a scenario including a centralized top UAV and a group of distributed bottom UAVs and obtained the optimal response delay of the closed-loop solution using random geometry and queuing theory. From the experimental simulation results, it can be seen that compared with the traditional algorithm without MEC, when this method is adopted, the total number of video streaming service packets transmitted by the drone to the data center is reduced by 89.9%. Messous et al. in [20] proposed a distributed algorithm with reference to the idea of game theory. In its proposed MEC service model, the UAV swarms offload high-intensity computing tasks to the base station through the wireless local area network and offload the high-intensity computing tasks to the edge servers with higher computing power than the base station through the cellular network. At the same time, a cost function of the weighted sum of energy consumption and delay is constructed as the system revenue index, and the energy consumption and execution delay are reduced by optimizing the unloading decision, so as to obtain the minimized cost function. Comparing the experimental results, it can be seen that the value of the cost function of this offloading decision is lower than that of only local computing, only offloading to the base station, and only offloading to the edge server. Judging from previous research results, the current UAV and edge computing architecture research has yielded certain achievements. However, with the continuous updating of application requirements, the traditional IP-based UAV network can no longer meet the development needs. In the traditional network model, the UAV as a network node is not only responsible for data forwarding but also for the exchange and transmission of routing information in the network. Due to the limitation of the UAV's own capabilities, the operation efficiency of complex routing protocols is also greatly weakened. Additionally, once the network has new business requirements, it will involve the specific settings and modifications of infrastructure-related functions. Based on this, we draw on the idea of SDN and set up an SDN controller in the network. The UAV platform is utilized as a switch and sensing node to collect contextual information and transmit it to the SDN controller, which reasonably controls and decides network functions and resource allocation based on network data information, so that UAVs in the network can be deployed more quickly and provide cloud services to users nearby.

As for the combination of SDN technology and mobile ad hoc networks, some research results deserve attention. Guo et al. [21] introduced SDN technology in the vehicle edge computing network to separate the control plane and data plane in the network. The SDN controller on the control plane is responsible for network traffic management and forwarding policy management, while the SDN switch on the data plane can perform functions such as data forwarding through the OpenFlow protocol according to the forwarding table and control commands sent by the control plane. Based on this, the author proposes an SDN-enhanced vehicle edge computing architecture, defines the optimal task offloading decision problem for minimizing delay, and proposes a task offloading scheme

based on the deep Q-learning algorithm. Through simulation experiments, the improved adaptability and performance of the proposed algorithm are verified. Mitsis et al. [22] studied the joint problem of end-user selection of MEC servers with optimal data offloading and optimal pricing of MEC servers in the context of multiple MEC servers and multiple end users. They introduced the SDN controller to execute the reinforcement deep learning framework based on the stochastic learning automata theory. The SDN controller is able to determine the reputation score of each MEC server and thereby influence the user's task offloading choices. Additionally, a low-complexity algorithm is introduced and designed to realize the data offloading and MEC server selection (DO-MECS algorithm). The optimal data offloading of the optimal MEC server is determined based on game theory and optimization techniques. The performance of the algorithm was evaluated through modeling and simulation under several scenarios, with both homogeneous and heterogeneous end users. Konstantinos et al. in [23] discussed a new generation of battlefield-oriented SDN-enabled ad hoc networks: SMANet. The network's ability to take advantage of SDN technology can help alleviate the inefficiencies of past decentralized mobile ad hoc designs. In addition, SDN can maintain the network flexibility and autonomy required for tactical operations. Importantly, SDN provides the means to easily configure and reconfigure nodes as strategies, mission objectives, and battlefield conditions change. Lastly, Zhao et al. [24] proposed an SDN-based optimization framework for UAV-assisted on-board computing offloading to minimize the system cost of on-board computing tasks. In this framework, UAV-hosted MEC servers can work on behalf of vehicle users, performing latency-sensitive and computing-intensive tasks. At the same time, UAVs can also be deployed as relay nodes to assist in forwarding computing tasks to the MEC server. The authors described the unloading decision problem as a multi-player computational unloading order game and designed a UAV-assisted Vehicle Computational Cost Optimization (UVCO) algorithm. Simulation results show that the proposed algorithm has better performance in both minimum average system cost (ASC) and system throughput. Compared with the previous studies, this paper is more suitable for the actual situation, since it considers the dependencies and parallelism of tasks before and after and proposes a better fine-grained task offloading scheme compared with the MEC combined with SDN technology. The deep reinforcement algorithm commonly used in the system has better performance. At the same time, combined with SDN technology, this paper sets up an SDN controller in the UAV network to realize the management and control of the entire network, and at the same time adopts an election mechanism to improve the survivability of the network.

### 1.3. Novelty and Contribution

Based on the task unloading problem of mobile intelligent users in urban environment, this paper proposes two task unloading strategies based on an improved EFO algorithm, which can be applied to user scenarios with different user sizes. In particular, sub-tasks face pre- and post-dependency and parallelism issues when performing fine-grained task offloading. Finally, the advanced nature of the algorithm is verified by evaluating and comparing the algorithm through the experimental platform. The main contributions of this paper can be summarized as follows:

- A UAV-assisted MEC system is constructed, and factors such as task dependencies, task transmission costs, and computational energy consumption are considered. The actual start and finish times of sub-task offloading to the UAV edge server are defined, and the computational offloading problem of the UAV-assisted MEC system is modeled.
- Combined with the constructed system model and relying on the network resource scheduling capability of SDN, two improved EFO algorithms are proposed for multi-mobile user scenarios in the urban environment.
- Based on the UAV platform, the performance of the two algorithms is compared through experiments. The comparison results show that the two task unloading strategies are superior in different application scenarios.

The remainder of this article is organized as follows. Section 2 mainly models the computational offloading problem of the UAV-assisted MEC system. Section 3 briefly introduces the basics of SDN and proposes improved centralized and distributed EFO algorithms based on system models. Section 4 simulates the algorithm based on the UAV platform. Section 5 draws a conclusion.

## 2. UAV Edge Network Modeling

Now, the combination of UAV and MEC computing is very extensive. In the traditional UAV auxiliary communication system, the UAV usually only transmits user data information as a communication relay. As shown in Figure 1, by combining MEC technology, UAV can also act as an MEC server to provide data processing services other than relay and forwarding for nearby ground users. When users are faced with data processing, the ground base station may be busy at this time and cannot process user data. Then, the user can choose the MEC server carried by the UAV as the unloading object. If the user data volume is large, the MEC server carried by the UAV can also upload the data to the ground base station after preprocessing, filter a lot of useless original data, and greatly reduce the bandwidth pressure of the ground base station. UAVs have the characteristics of strong mobility, flexible deployment, energy-saving, and wide coverage. It can break through the application limitation of traditional ground-station-based MECs and provide technical solutions for mobile communication in urban environments.

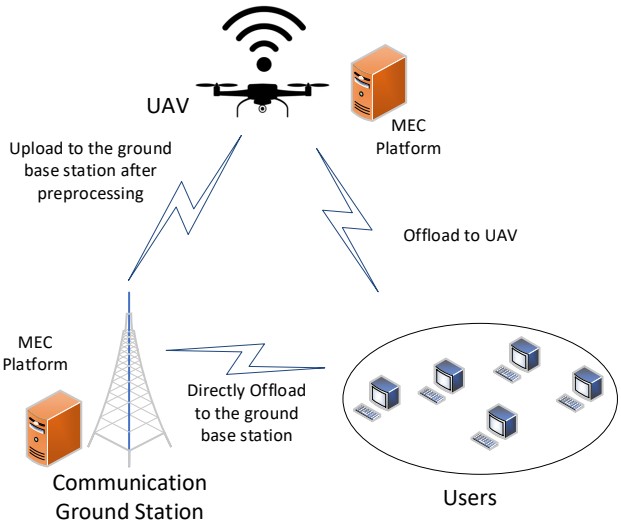

**Figure 1.** Single-UAV edge platform.

### 2.1. System Model

In the UAV edge network, we consider a set of mobile devices $n \in \{1, 2, \ldots N\}$, and the MEC server $k \in \{1, 2, \ldots K\}$ is equipped and mounted on the UAV platform at the same time. Based on the model, in time slot $i\epsilon\{1, 2, \ldots, I\}$, the number of sub-tasks is represented by $m \in \{1, 2, 3 \ldots M\}$, the coordinate position of the mobile user is set to $U_n(i) = [x_n(i), y_n(i)]^T \in R^{2 \times 1}$, the initial position of the UAV is $U_k(i) = [x_k(i), y_k(i)]^T \in R^{2 \times 1}$, the final position is $U_k(i+1) = [x_k(i+1), y_k(i+1)]^T \in R^{2 \times 1}$, and the ground height is $H$.

In the process of data transmission, even if there are multiple available MEC servers based on the UAV platform, the single-user equipment normally can only select one of them. Meanwhile, when the user device offloads the task, the nearest mobile edge server is not always the best solution because even the closest distance cannot avoid the congested communication environment and excessive cost of a mobile edge server, which fails to fulfill the requirements of the task. Therefore, various factors should be considered to decide whether to process locally or on the MEC server on the UAV platform.

This paper focuses on fine-grained task offloading, which is different from binary task offloading, which takes the entire mobile terminal application as the offloading object. However, in practice, we usually divide the task into multiple sub-tasks with pre- and post-dependency and parallelism. These divided sub-tasks consume fewer computing resources and communication transmission resources. Therefore, some or all tasks can be offloaded to multiple remote servers for processing, thereby saving computing time and transmission time and also achieving higher resource utilization of edge server clusters.

When this type of task uninstallation is divided into multiple sub-tasks due to application tasks, there is a partial order relationship between them, which is generally expressed by a directed acyclic graph (DAG): $G = (T,E,D)$, where $T = \{T_1, T_2, \ldots, T_n\}$ is a set of sub-tasks, $E$ represents a directed edge set of dependencies between sub-tasks, and $D$ represents the data size of the sub-task. The task dependency of the application service is defined as: if $Tj$ must be run after $Ti$ is executed, there is a directed edge from $Tj$ to $Ti$, with $Ti$ being the predecessor task of $Tj$, and $Tj$ being the successor task of $Ti$. The weight of each edge indicates the data flux between tasks. At the same time, $Tj$ and $Tl$ are parallel tasks, as shown in Figure 2. In order to increase the fine granularity of tasks, many task partitioning methods have been applied. For example, recursive division, function division, data division, detection division, etc. These methods solve the display division of computing tasks on a macro level. On the micro level, common methods include graph division, hypergraph division, etc., which can further divide tasks.

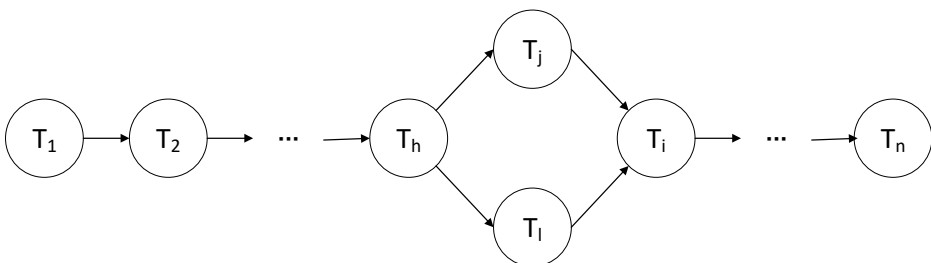

**Figure 2.** Task DAG graph.

In a UAV edge network system, $T_{n,m}$ is used to represent the $j$th sub-task of device $i$, and it can be described by three physical quantities: $T_{n,m} = (d_{n,m}, c_{n,m}, a_{n,m})$, where $d_{n,m}$ means the size of the input data, $c_{n,m}$ the CPU cycle completed by the task, and $a_{n,m}$ the uninstallation decision of the $m$th task of the device $n$. When $a_{n,m}$ is 0, the computing task will be calculated locally; when its value is 1, the computing task will be offloaded to the UAV edge server for processing.

### 2.2. Problem Formulation

In this paper, a task offloading system based on an improved EFO algorithm is proposed on the basis of [25], which mainly solves the problems of task offloading priority and offloading decision making. First, we systematically model the UAV-assisted MEC model.

Due to the line-of-sight communication between the UAV and the mobile user, the channel power gain [26] can be expressed as:

$$g_{n,k}(i) = \frac{\beta_0}{H^2 + ||J(i+1) - Un(i)||^2},$$ (1)

where $\beta_0$ represents the channel power gain at a distance of 1 m. Therefore, the information rate of the UAV and mobile user $n$ in the $i$th time slot is:

$$r_{n,k}(i) = Blog_2\left(1 + \frac{P_n g_{n,k}(i)}{\sigma^2 + P_L f_L(i)}\right),$$ (2)

Among them, $B$ represents the channel bandwidth, $P_n$ represents the transmit power in the upload link of the mobile user $n$, $\sigma^2$ represents the noise power, and $P_L$ represents the transmission loss of the UAV. $f_L(i)$ is the barrier coefficient between the drone and the mobile user, and its value rule is:

$$f_L(i) = \begin{cases} 1 & Lossinair \\ 0 & Nolossinair \end{cases}' \tag{3}$$

For the energy of UAV flight, since this paper assumes that the UAV is in a constant altitude flight state, the calculation of UAV flight energy only considers the kinetic energy of the UAV, does not consider the gravitational potential energy, and is only related to the speed of the UAV. UAV flight energy can be expressed as:

$$e_{fly,k}(i) = \lambda \|v_k(i)\|^2, \tag{4}$$

where $\lambda = 0.5Mt_{fly}$, $M$ is the load factor of the UAV, and $v_k(i)$ shall not exceed the maximum speed $V_{\max}$ of UAV in numerical value. Next, we model the latency and energy in the system into two parts, local and edge.

Local computing: The latency for the local device to process the task is:

$$t_{l,n,m} = c_{n,m} / f_n^l, \tag{5}$$

where $f_n^l$ represents the computing capability of the local user equipment. The energy consumption of the local processing task is:

$$e_{l,n,m} = \varepsilon c_{n,m} \tag{6}$$

where $\varepsilon$ represents the computing energy consumption per unit CPU of the local user equipment.

Edge side: On the edge side, the delay processed by the MEC server carried by the UAV includes transmission delay, queuing delay, and task processing delay:

$$t_{k,n,m}(i) = d_{n,m} / r_{n,k}(i) + t_{n,m}^q + c_{n,m} / f^k \tag{7}$$

where $t_{i,j}^q$ represents the queuing time of sub-tasks, and $f^k$ represents the computing power of the MEC server. The energy consumption for edge-side processing is:

$$e_{tr,n,m}(i) = P_n d_{n,m} / r_{n,m}(i) \tag{8}$$

$$e_{k,n,m} = \varepsilon_k c_{n,m} \tag{9}$$

where $\varepsilon_k$ represents the energy consumed by the unit CPU of the MEC server. From the previous section, we know that the tasks of the device application are represented by a DAG, and certain priority exist in these sub-tasks. First of all, for tasks with front and back dependencies, the predecessor sub-task has a higher priority over the successor sub-task; Secondly, there are priorities among parallel sub-tasks, which require to be quantified and compared by specific values.

As shown in the Figures 3 and 4, we compared the total delay of two kinds of random scheduling and scheduling according to priority. The priority relationship of sub-tasks $a$, $b$, $c$, and $d$ in the figure is $a > b > c > d$. It can be seen from the comparison that the task execution must meet the requirements that the task with high priority can be calculated only after the task with low priority is calculated. Therefore, in Figure 3, there is a time gap between computing time nodes of different tasks, so the total time delay of random scheduling is far greater than the total time delay of scheduling according to priority, so it is necessary to determine the priority of sub-tasks.

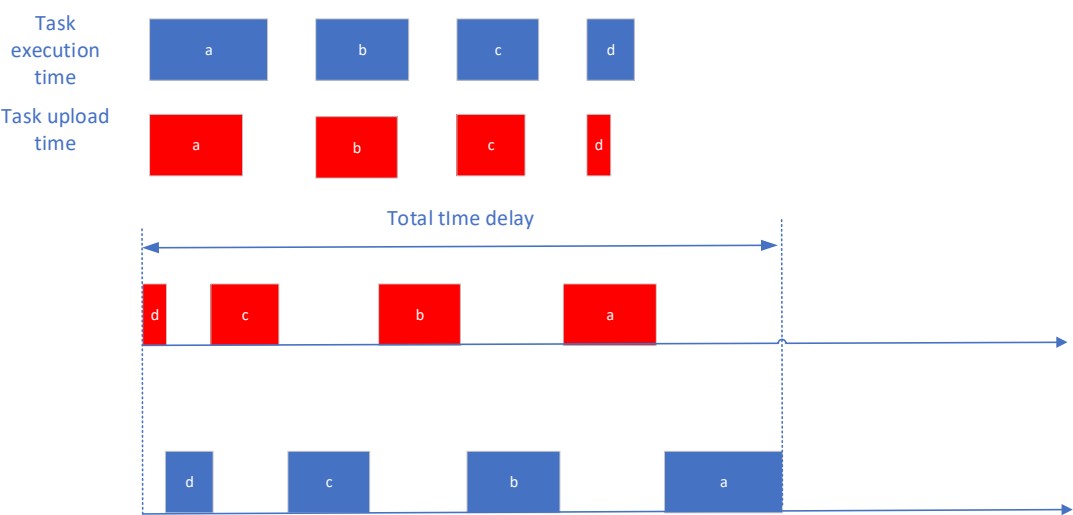

**Figure 3.** Schematic diagram of random scheduling.

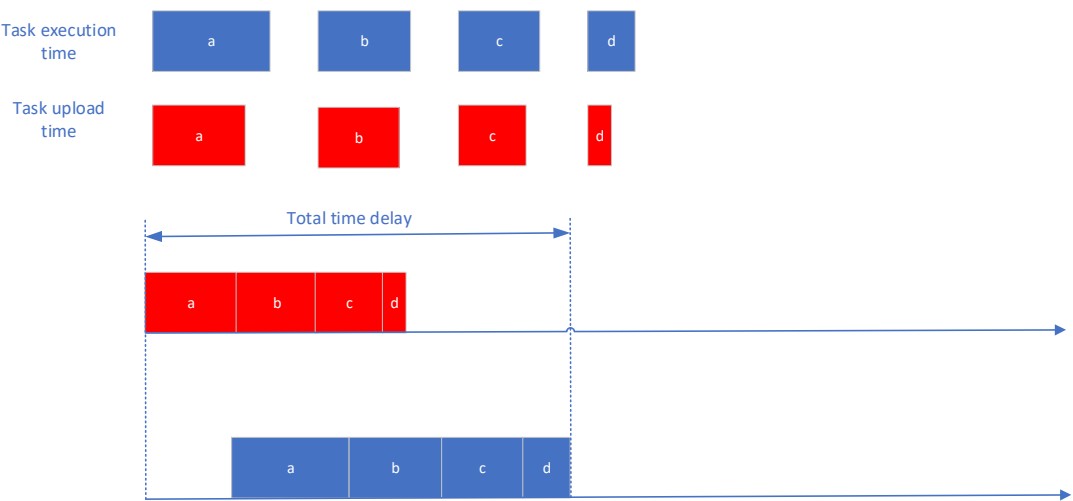

**Figure 4.** Schematic diagram of scheduling according to priority.

To quantify the priority, we define the communication cost of sending data from sub-task $m'$ to sub-task $m$ as:

$$c_{m',m}(i) = \begin{cases} 0 & if\, a_{n,m} = a_{n,m'}, \\ data_{m',m}/r_{n,k}(i) & oterwise, \end{cases} \tag{10}$$

From Formula (10), $data_{m',m}$ represents the amount of data that sub-task m' sends to sub-task $m$. When the sub-tasks $m'$ and $m$ are scheduled on the same processor, $c_{m,m'}$ are zero, because we assume that the communication cost within the processor is negligible. The communication cost of sending data from sub-task $j'$ to sub-task $j$ is:

$$\overline{t_{m,m'}(i)} = \overline{c_{m,m'}(i)} = data_{m',m}/(2r_{n,k}(i)), \tag{11}$$

The average execution cost of sub-task $j$ is defined as:

$$\overline{t_{k,n,m}(i)} = \left(t_{k,n,m}(i) + t_{l,n,m}\right)/2, \tag{12}$$

Among them, $\omega_{n,m}^{k}$ is the average execution time of sub-task $j$, $t_{n,m}^{l}$ indicates the time processed locally, so we represent the priority quantity as:

$$rank(m) = \overline{t_{k,n,m}(i)} + \max_{m' \in succ(m)} \left( \overline{t_{m,m'}(i)} + rank(m') \right), \tag{13}$$

Among them, $Succ(m)$ represents the set of successor tasks of task $m$, and $rank(m)$ is the length of the critical path from task $m$ to the *exit* task, which is proportional to the priority of the task and is an important indicator to measure the priority of the task.

Second, we also need to consider how to minimize the average task time to respond to all users in the UAV edge network. For this, we adopt $AST(m,k)$ and $AFT(m,k)$ to, respectively, denote the actual start time and finish time of offloading sub-tasks to the UAV edge server.

Before calculating $AST(m,k)$ of task $m$, the predecessor sub-task before task j must be fully scheduled to calculate its value recursively, so the calculation formula of $AST(m,k)$ is as follows:

$$AST(m,k) = \max \left\{ avail\{0 \cup [k]\}, \max_{m' \in pred(m)} \left( AFT(m') + c_{m,m'} \right) \right\} \tag{14}$$

where $pred(m)$ is the set of sub-tasks that are the direct predecessors of sub-task $m$. $C_{m,m'}$ represents the communication cost from $m'$ to $m$, which is mainly caused by the transfer of task $m$ to task $m'$. What is more, $avail\{0 \cup [k]\}$ is the earliest time that the server or the local device is ready to execute the task under the task-dependent condition. The actual execution completion time formula is illustrated as follows]:

$$AFT(m\prime) = \min \left\{ \omega_{n,m}^{k\prime} + AST(m\prime, k\prime) \right\} k\prime \in \{0\} \cup \{k\}, \tag{15}$$

In Formula (16), $\omega_{n,m}^{k}$ is the execution time either on the UAV edge server or on the local CPU. After all the sub-tasks in the DAG of the task are scheduled, the execution time of the application will be calculated as the actual completion time of the exit task (tasks without sub-tasks are called exit tasks):

$$AFT_n = AST(exit) + t_n^{last}, \tag{16}$$

Among them, $t_n^{last}$ is the execution time of the last task, which is generally responsible for collecting execution results, so it is usually executed locally. Therefore, the task offloading problem can be expressed as:

$$
\begin{cases}
\min \left( \sum_{i=1}^{I} \sum_{n=1}^{N} \sum_{m=1}^{M} AFT_n(i) \right) & (Obj) \\
s.t \; E_n \geqslant \sum_{n=1}^{N} \sum_{m=1}^{M} X_m e_{lo,n,m} + \sum_{i=1}^{I} \sum_{n=1}^{N} \sum_{m=1}^{M} (1 - X_m) e_{tr,n,m}(i) & (C1) \\
E_k \geqslant \sum_{n=1}^{N} \sum_{m=1}^{M} (1 - X_m) e_{k,n,m} + \sum_{i=1}^{I} \sum_{n=1}^{N} \sum_{m=1}^{M} e_{fly,k}(i) & (C2) \\
\{x_n(i) \in [0, L], y_n(i) \in [0, W]\}, \forall i, n & (C3) \\
\{x_k(i) \in [0, L], y_k(i) \in [0, W]\}, \forall i & (C4) \\
\eta_n(i) \in \{0, 1\} \forall i, n & (C5)
\end{cases} \tag{17}
$$

where $X_m$ indicates whether the sub-task is processed locally. When it is processed locally, its value is 1, and when it needs to be uploaded to the MEC server, its value is 0. Thus, Equation (17) indicates that after traversing the unloading decision of all user sub-tasks, the objective function with the minimum actual completion time that can be applied in the application scenario is selected. Constraints *C1*, *C2* are the energy consumption constraints of intelligent terminals and UAVs; Constraints *C3*, *C4* are the moving range of UAV and intelligent terminal; and Constraint *C5* is the value range of blocking coefficient between UAV and intelligent terminal. It can be seen from the expression that this paper only considers the situation of unloading tasks to one UAV for the time being, which will be

expanded to multiple UAV scenarios in the following improvements. According to the above model, the program diagram of the EFO algorithm is as shown in Figure 5:

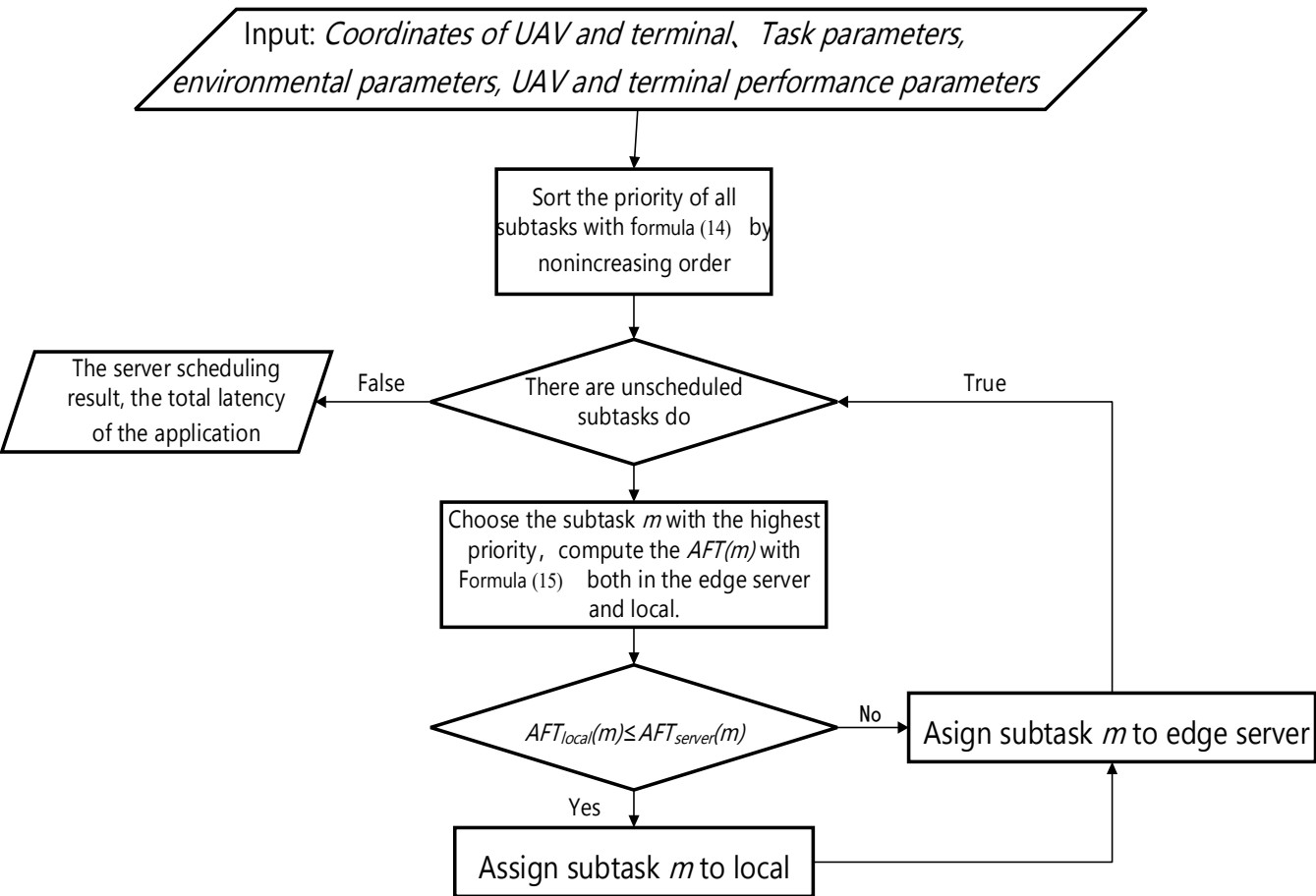

**Figure 5.** Program diagram of the EFO algorithm.

In the EFO algorithm, the task unloading strategy of the sub-task is analyzed mainly according to the data volume of the sub-task, the task topology, and the computing power of the MEC server and the local processor. It can be concluded from the above algorithm steps that the algorithm first sorts the priorities of the sub-tasks, and then successively selects the highest priority and unassigned sub-tasks for unloading decision. The judgment basis is to compare the local processing time with the processing time of the MEC server and finally output the unloading decision of the sub-task.

## 3. UAV Edge Network

### 3.1. Architecture Pattern

The UAV edge network is shown in Figure 6. The UAV edge network can serve a wide range of urban environments and can also serve multiple Desired Regions (DRs) in the city, such as suburbs, business districts, school districts, etc. Each UAV will stay for a period of time when passing the DR. During this period, the terminal in the DR uploads the data to be processed to the UAV. The UAV processes the data according to the trained model and then returns it to the terminal to achieve better edge computing services.

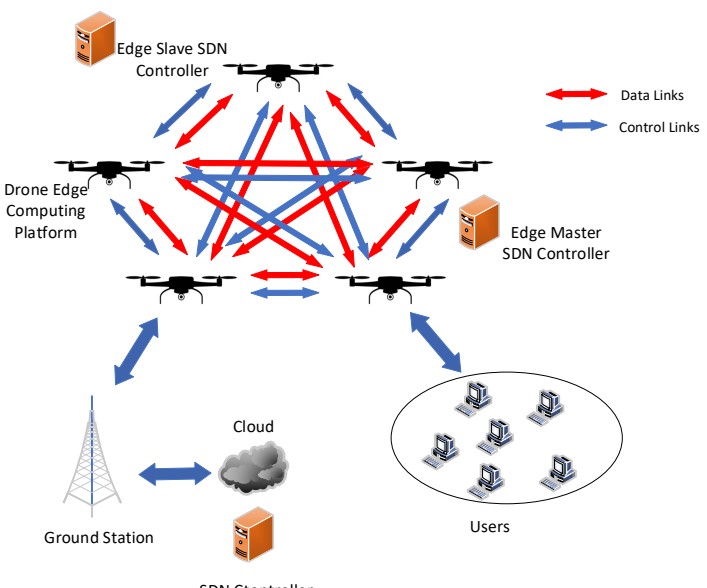

**Figure 6.** UAV edge network based on SDN architecture.

In the UAV edge network, we set the SDN controller to sense the network topology and realize the scheduling between different user tasks to simplify the functions of the UAV equipment in the network. The advantages of introducing SDN technology into the architecture are:

- SDN borrows the ideas of software definition to realize the programmable UAV edge network, which makes the network more flexible and open, the deployment of new applications and services more convenient, and the network management simpler.
- The SDN controller collects and processes network-wide information and pools and shares the resources of the UAV platform and user equipment, which facilitates comprehensive planning and management of resources, thus making better decisions in terms of resource scheduling, load balancing, and path selection.

The SDN architecture can meet various demands in the UAV edge network, including supporting flexible routing switching strategies, handling unstable communication links, and supporting node mobility. We refer to [27] to use the election mechanism to elect the cluster head and cluster members and set the corresponding SDN controller in the UAV swarm. The difference between the two is that the SDN controller of cluster members generally does not enable communication management, formation management, and security management functions on the application plane. Therefore, in the network, cluster members are mainly responsible for information collection, and the cluster head is responsible for information uploading. Based on this, the cluster head can give the SDN controller a global view to better manage the structure of the network, always ensure that there is a command and decision center in the network, and ensure the efficiency of task execution. At the same time, since each UAV in the cluster is equipped with an SDN controller with the same internal components and structure, the election mechanism can support the migration of cluster heads between UAVs. Once a task is required, or the master SDN controller is paralyzed, other slave SDN controllers can act as the master controller by turning on certain components and functions.

Based on the above structure, we set up the MEC server on the UAV to provide better computing services for end users. Moreover, based on the SDN controller carried by the UAV, the UAV needs to continuously exchange its own coverage information and location information, so as to achieve the integration of the entire network and full coverage of the area. At the same time, we set up an SDN controller in the data center to control and manage the UAV edge network based on the ground network. Usually, the UAV equipped with the MEC server is set at the edge of the network at a communication distance of

one or two hops away from the user, so as to cover a certain range of users, which can effectively facilitate computing offloading for users within the coverage area. At the same time, this paper integrates the concept of Network Functions Virtualization (NFV) into the MEC server to virtualize the computing and storage resources of the MEC server with the purpose of carrying the functions of different applications and services. The SDN control module also runs on the NFV-enabled MEC server and utilizes network global information to efficiently allocate virtualized resources. Moreover, each MEC server can upload its own state information, including data uploaded from different users and QoS requirements, to the cloud SDN controller after preprocessing. Based on the received information, the SDN controller then makes decisions on task migration between MEC servers [28].

### 3.2. Algorithm Construction

### 3.2.1. Centralized EFO Algorithm

In a traditional multi-user service scenario, since a single user cannot perceive the offloading of other users when performing task offloading, policy information will not be propagated among users, which is not conducive to task scheduling. The SDN-based UAV network proposed in this paper uses the settings of the SDN controller to achieve logical centralized control of the devices in the network.

Each user equipment in the UAV network uploads its own task topology and local processing capabilities to the SDN controller before when and where to schedule the subtasks of different users is decided by the SDN controller. The program diagram is shown in Figure 7.

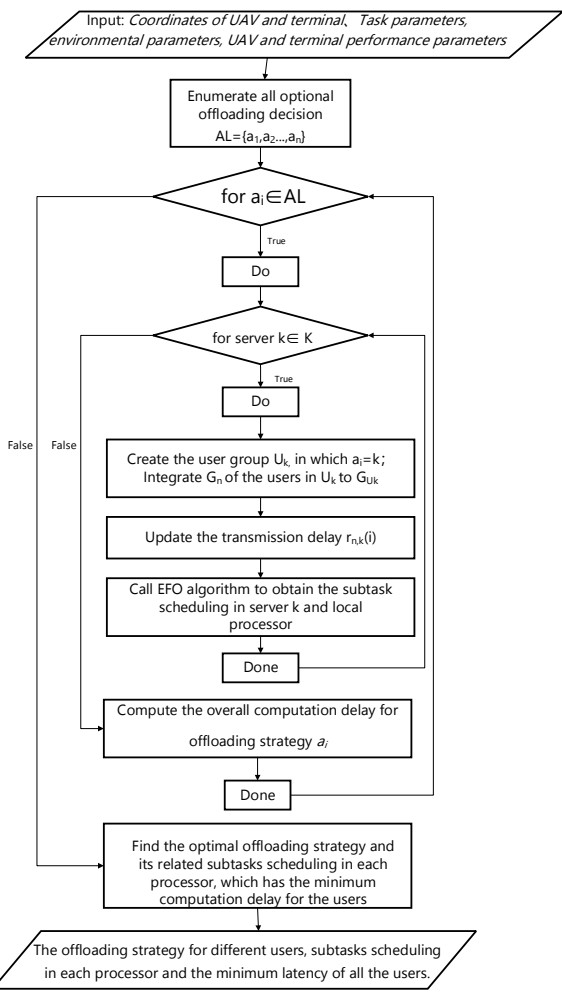

**Figure 7.** Program diagram of the CEFO algorithm.

The above centralized EFO algorithm (CEFO) demonstrate that all offloading decisions made by users in the UAV edge network are enumerated first, and the graphs of user tasks that are offloaded to the same UAV edge server are then merged into an integrated DAG graph. Where AI represents the user uninstall option that can be referenced.

Finally, after calculating the processing delay of the "total DAG", the offloading scheme with the lowest latency is selected as the offloading strategy. However, with the increase in the number of network users, the convergence time of the algorithm also increases, so it is difficult to be applied to network scenarios with a large amount of users.

### 3.2.2. Distributed EFO Algorithm

Based on the above discussion, we use the idea of "game theory", which means that when multiple users perform task offloading, one user's equipment makes a computing offloading decision to respond to the previous decision-making behavior of other participating users in each step and makes a local optimal decision. After a limited number of steps, all users can, through self-organization, achieve a relatively balanced state, which is called Nash equilibrium, and a DEFO algorithm is introduced. The program diagram is shown in Figure 8.

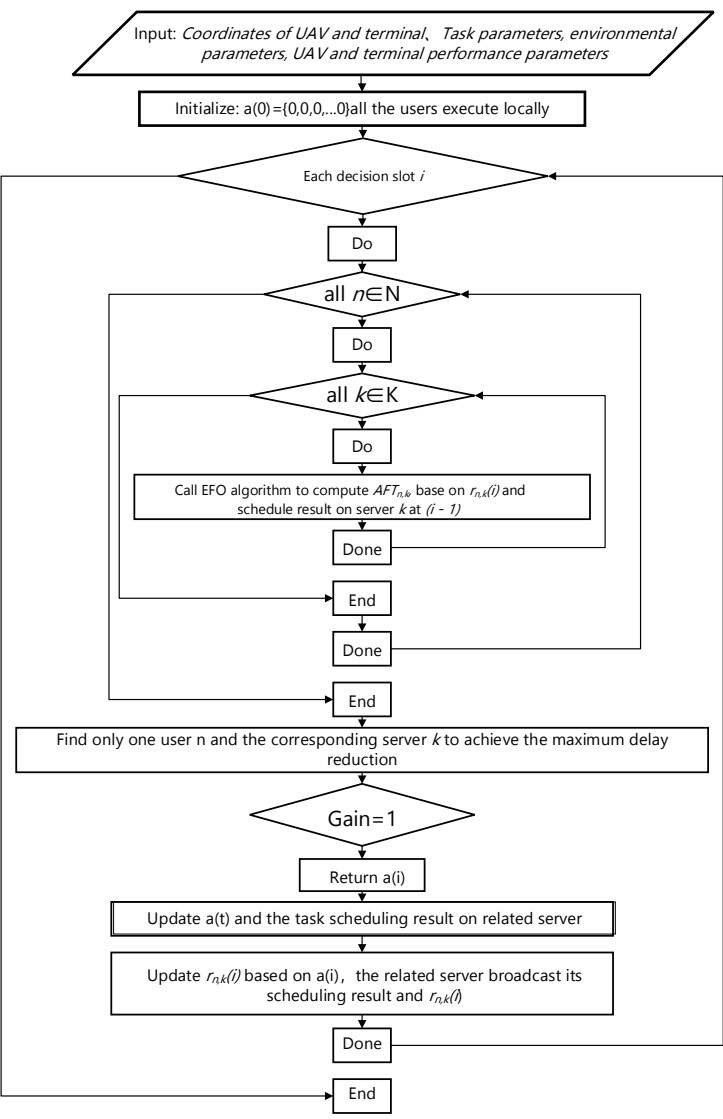

**Figure 8.** Program diagram of the DEFO algorithm.

As shown in Figure 9, there are three devices and two UAV-assisted MEC servers in the UAV network. In the initialization step, each device executes all its sub-tasks locally and calls the EFO algorithm in the first iteration to calculate the AFT value. In the iterative process, assuming that the wireless channel is free from interference, no other sub-tasks are performed on the UAV edge server. Then, the controller will select user3 to unload their sub-tasks on UAV server2, because this action generates lower latency compared with the local execution time; after that, user3 executes the EFO algorithm and offloads sub-task2 and sub-task4 to UAV server2. Next, UAV server2 uploads the scheduling result and its wireless channel rate ri,k(α) to the SDN controller, and the information is then broadcasted to all devices in the UAV network to facilitate the further choice-making process for the user in the next iteration. In Figure 9c, user1 uploads sub-task3 to UAV server1 according to the previous broadcast information to further reduce the latency value. Through N times of iterations, each user achieves the Nash equilibrium. In the final equilibrium state (Figure 9d), it is impossible for each single user to unilaterally change the strategy to further reduce its calculation latency. What is more, if Gain = 1, the DEFO algorithm is terminated.

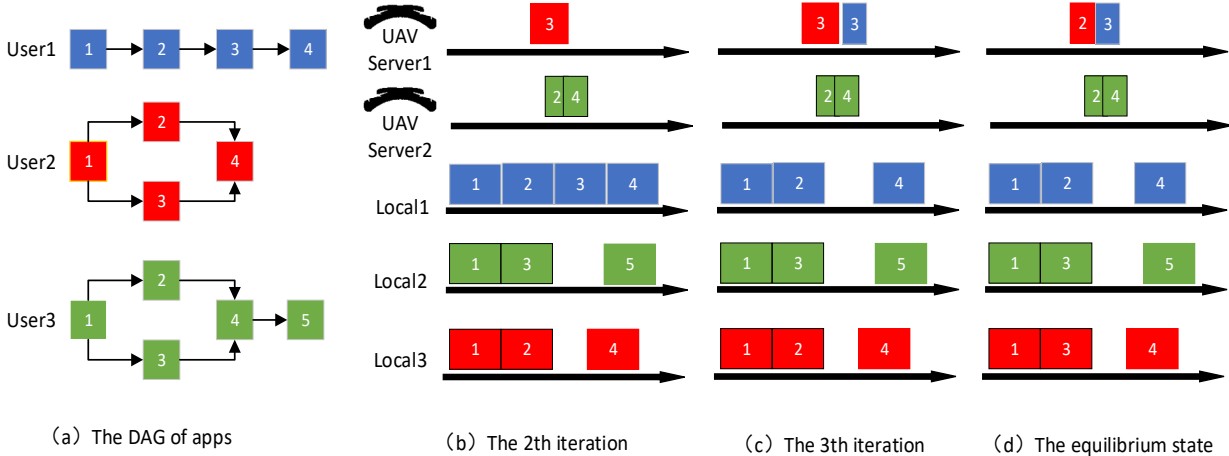

**Figure 9.** Example of DEFO.

It can be seen that through the above DEFO algorithm that each UAV network device is no longer an independent individual when performing task offloading. Instead, it comprehensively considers the information of other devices during the uninstallation process and adjusts its uninstallation strategy accordingly. When no more adjustments are made, the final equilibrium state is reached, and the latency is minimized at this time.

### 3.2.3. Computational Complexity

In this section, we discuss the time complexity of the algorithm, which represents the time cost of the algorithm. In the CEFO algorithm, on the basis of traversing all UAV edge servers, the mobile user needs to obtain the minimum average user unloading delay in the system considering different unloading strategies. Therefore, the algorithm is exponentially related to the number of mobile users and the number of UAV edge servers.

First, set the number of mobile users in the scene to $N$ and the number of UAV edge servers to, then since each mobile user needs to traverse all UAV edge servers, there are:

$$\binom{0}{M} \cdot N^0 + \binom{1}{M} \cdot N^1 + \ldots \ldots \binom{M}{M} \cdot N^M = \sum_{i=0}^{p} \binom{p}{M} \cdot N^p = (N+1)^M, \quad (18)$$

where $p$ represents the number of tasks offloaded to the UAV edger. Since each offloading strategy also needs to call the EFO algorithm, and EFO itself is an algorithm that selects the offloading strategy in the order of priority, and its complexity is $O(n+e)$, where $n$

represents the number of nodes, and *e* represents the number of directed edges in the DAG. So, the complexity of the CEFO algorithm is $O\left[(N+1)^M \cdot (n+e)\right]$.

The principle of the DEFO algorithm is that each user makes decisions by referring to the decisions of other users in the previous stage, without having to traverse every UAV edge server, so the complexity is greatly reduced. Because only one user can update the policy by analyzing the unloading decisions of other users in each iteration, the algorithm complexity is independent of the user size, so the complexity of the DEFO algorithm is $O(1)$.

## 4. Experimental Evaluation and Comparison

In this section, we conduct an experimental evaluation of the algorithm and examine the performance characteristics of the algorithm based on the application background of UAV edge network. The main consideration of this paper is the multi-user MEC scenario of the UAV edge network. In this network architecture, a certain number of UAV platforms are randomly distributed in space, and a certain number of heterogeneous edge servers are set. The server is usually located near the user who is able to choose a variety of ways to process data when needed, and the latency can be minimized. With the combination of application background, we set up the following experiment.

According to the literature [29], the communication environment we set is: transmission power of 100 mWatts, background noise of 100 dBm, and wireless channel bandwidth of 20 MHZ. Additionally, the CPU frequency of the edge server (500 MHZ) is 10 times that of the user equipment (5000 MHZ). At the same time, we randomly generate some DAGs, the size of their sub-tasks and the required CPU cycles range being between [500, 1500] KB and [0.2, 0.3] GHZ, respectively.

### 4.1. Experimental Evaluation

As shown in Figure 10 and Table 1, the UAV we use mainly includes four modules: GPS module, Kerloud PX4 flight controller, CSI camera, and NIVIDA Jetson Nano in this paper. Among them, the NVIDIA Jeston Nano is installed on some UAVs as an embedded edge server. Instead of being equipped with any deep-learning-specific accelerators, it is mainly installed with NVIDIA Maxwell GPU, which also has good computing power and can reach an overall peak performance of 472 GFLOPs. At the same time, the weight of the processor is about 140 g, the physical size is only 69 mm × 45 mm, and the price is moderate. Furthermore, it provides a variety of standard interfaces, including Gigabit Ethernet port, HDMI port, USB 3.0 port (×4), and so on. Therefore, this paper considers the NVIDIA Jetson Nano as a more suitable edge server mounted on the UAV to process the data offloaded by the user. Because the maximum load of the UAV can reach 1.2 kg, while the weight of the NIVIDA Jetson Nano module is only 140 g, its computing power is 5 to 10 times the average computing power of the general terminal, which is enough to provide reliable computing services.

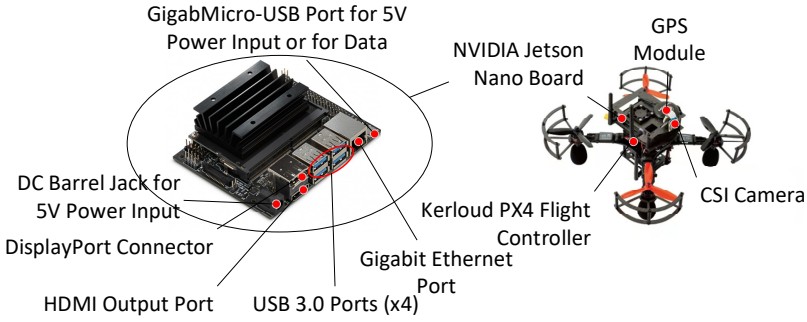

**Figure 10.** UAV with NIVIDA Jetson Nano and NIVIDA Jetson Nano interface schematic.

**Table 1.** NVIDIA Jetson Nano Performance Metrics.

| Title 1 | Title 2 |
| --- | --- |
| GPU | 128-core Maxwell |
| CPU | Quad-core Arm A57 @ 1.43 GHz |
| Memory | 4 GB 64-bit LPDDR4x at 25.6 GB/s |
| Peak performance | 472 GFLOPs |
| Native precision support | FP16/FP32 |
| Storage | Micro SD card slot or 16 GB eMMC flash |
| Operating temperature | Ranges between −25 and 80 Celsius |
| Power consumption | 5 W/10 W |
| Physical dimensions | 69 mm × 45 mm |
| Weight | 140 g |
| Price | USD 99 |

### 4.2. Algorithm Characteristics

The experiment first examines the characteristics of the algorithm in a given application context. As shown in Figure 11, the DAG of each task is composed of 10 sub-tasks. It can be observed from the Figure that when the number of users increases to 50, the server resource utilization rate reaches a peak of 64%. This experimental result has reference value for the setting of MEC servers with a specific number of users.

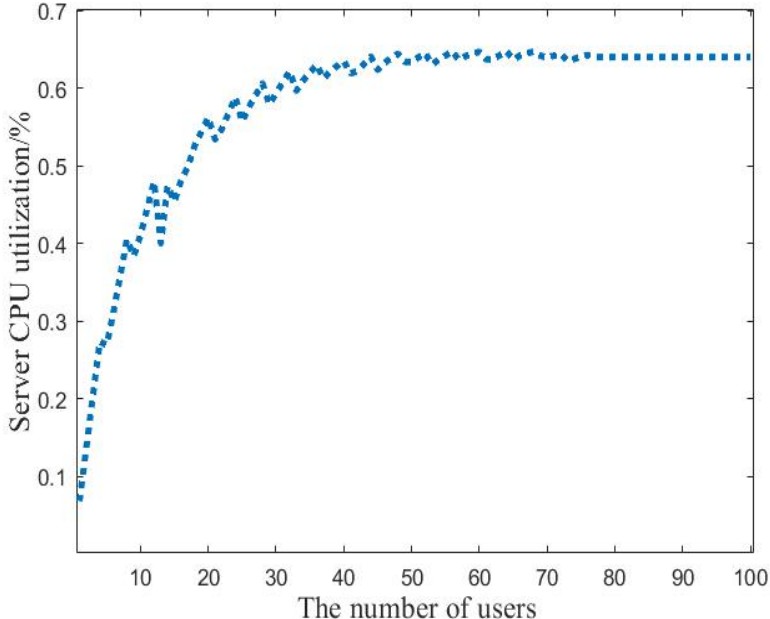

**Figure 11.** Relationship between the number of users and server resource utilization.

In the experiment, 30 users and 4 servers are set to explore the relationship between sub-tasks and CPU utilization, as shown in Figure 12. It can be seen from the figure that when the number of sub-tasks of user terminals in the multi-UAV system increases from 2 to 100, the resource utilization of the server is also improved to a certain extent. When the number of sub-tasks reaches a certain number, the network will reach saturation. Therefore, it can be inferred that in a network with a certain number of users, the server utilization can be further improved by dividing the application into more sub-tasks (in a more fine-grained way). More importantly, the average delay can be reduced by fine-grained task partitioning, as shown in Figure 13.

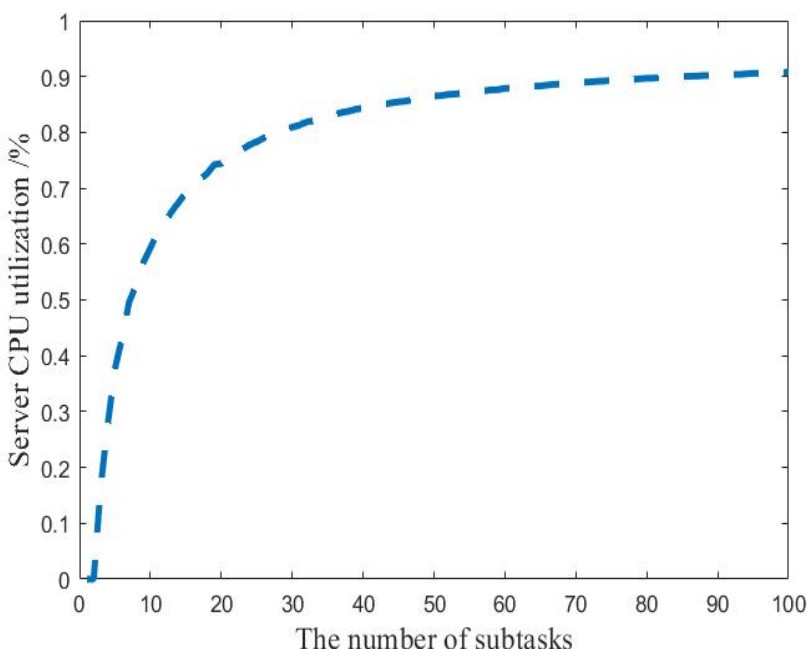

**Figure 12.** Relationship between sub-tasks and CPU utilization of servers.

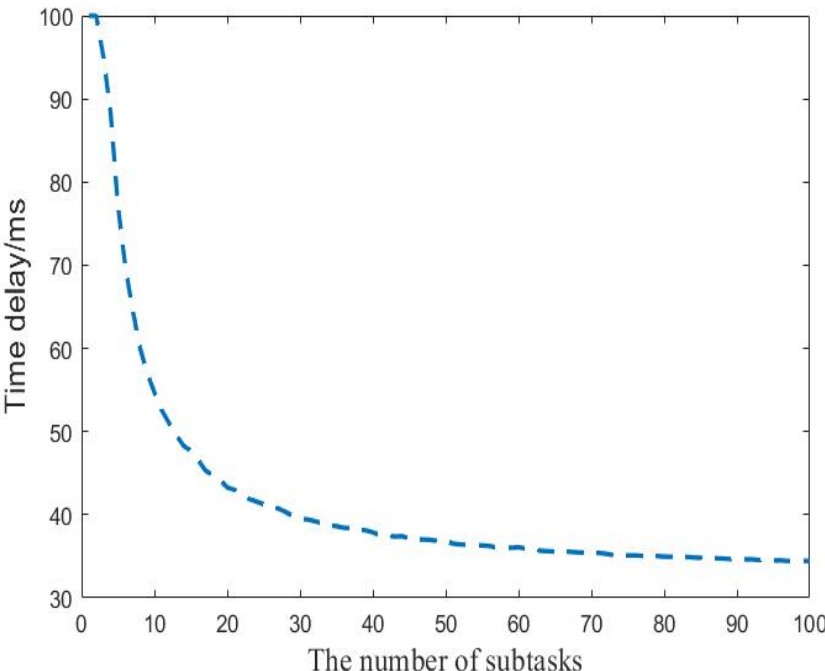

**Figure 13.** Relationship between sub-tasks and average delay.

In order to further examine the influence of the number of users and the number of servers on the convergence time in the DEFO algorithm, we set up 50 and 100 users equipped with 5 and 10 MEC servers, respectively. We can observe from Table 2 that the DEFO algorithm can reach the Nash equilibrium after a finite number of iterations under different conditions. Furthermore, what can be deduced from the results is that the convergence time will increase with the growth in the number of users, but the rise in the number of MEC servers will help reduce the average latency of users in the network.

**Table 2.** Relationship between number of users, number of servers, and convergence time.

| User number | Number of servers | Number of iterations | | | | | | | | |
|---|---|---|---|---|---|---|---|---|---|---|
| | | 1 | 5 | 10 | 15 | 20 | 25 | 30 | 35 | 40 |
| | | Time delay/ms | | | | | | | | |
| 50 | 5 | 245.8 | 233.4 | 219.8 | 208.8 | 202.2 | 200.8 | 197.4 | 196.1 | 196.7 |
| | 10 | 240.6 | 226.9 | 211.0 | 201.0 | 188.3 | 179.1 | 173.1 | 168.7 | 166.7 |
| 100 | 5 | 253.1 | 246.4 | 238.3 | 232.3 | 228.9 | 227.0 | 221.5 | 220.5 | 220.0 |
| | 10 | 250.7 | 244.0 | 236.8 | 230.0 | 224.6 | 218.8 | 213.5 | 208.3 | 206.7 |

| Number of users | Number of servers | Number of iterations | | | | | | | | |
|---|---|---|---|---|---|---|---|---|---|---|
| | | 45 | 50 | 55 | 60 | 65 | 70 | 80 | 90 | 100 |
| | | Latency/ms | | | | | | | | |
| 50 | 5 | 194.7 | 192.7 | 196.7 | 196.8 | 195.1 | 195.4 | 196.7 | 196.8 | 195.4 |
| | 10 | 161.0 | 159.7 | 158.0 | 158.0 | 158.0 | 158.0 | 158.0 | 158.0 | 158.0 |
| 100 | 5 | 217.0 | 216.0 | 215.7 | 215.6 | 215.0 | 212.9 | 214.0 | 217.8 | 215.3 |
| | 10 | 203.3 | 200.2 | 198.6 | 194.8 | 195.5 | 194.8 | 192.0 | 190.1 | 191.0 |

*4.3. Comparative Experiment*

Referring to the experimental results, we compare the centralized algorithm with the distributed algorithm and refer to the PGOA algorithm in [30], which is also applied to a distributed task offloading system with multiple users and multiple servers.

First, we compare the running time and time delay of the DEFO algorithm and the CEFO algorithm by setting up two servers and 1 to 8 user devices in the network. According to the results in Tables 3 and 4, the two algorithms have their own advantages. In terms of computational complexity and computational cost, compared with the DEFO algorithm, the running time of the CEFO algorithm increases exponentially with the number of users, and the computational complexity and computational cost are higher, while the DEFO algorithm has only one user per iteration. The uninstall decision of other users is used to update the policy, so the runtime is independent of the number of users. In terms of algorithm delay, with the increase in users, the delay of the CEFO algorithm is slightly lower than that of the DEFO algorithm. Based on the above results, it is concluded that although the CEFO algorithm reduces the delay better, the more users, the more scale of the terminals that need to be traversed, and the more running cost of the time. Therefore, the CEFO algorithm is difficult to apply to the UAV edge network, which involves a large number of users.

**Table 3.** Relationship between number of users and running time.

| Running Time/ms | Algorithm Type | |
|---|---|---|
| Number of Users | CEFO | DEFO |
| 1 | 0.072 | 0.0072 |
| 2 | 0.05 | 0.05 |
| 3 | 0.01 | 0.01 |
| 4 | 0.02 | 0.02 |
| 5 | 1.139 | 0.032 |
| 6 | 8.733 | 0.03 |
| 7 | 69.79 | 0.016 |
| 8 | 717.7 | 0.031 |

**Table 4.** Diagram of latency comparison.

| Number of Users ⟍ Latency/ms | Algorithm Type | |
|:---:|:---:|:---:|
| | **CEFO** | **DEFO** |
| 1 | 94.0 | 94.0 |
| 2 | 94.0 | 94.0 |
| 3 | 102.3 | 105.0 |
| 4 | 109.3 | 111.5 |
| 5 | 118.5 | 120.8 |
| 6 | 124.0 | 125.3 |

Then, we compare the average latency of the DEFO algorithm and the PGOA algorithm. Figure 14 demonstrates the average task latency of edge servers under different network sizes. Here, 20, 30, and 50 users are set up in the network, respectively. It can be seen that with the increase in the number of servers, the average latency of the two algorithms is on the decrease; with the rise in the number of edge servers from 2 to 20, the average latency achieved by the DEFO algorithm is much lower than that of the PGOA algorithm. At the same time, the number of users plays a significant role in the increase in time delay.

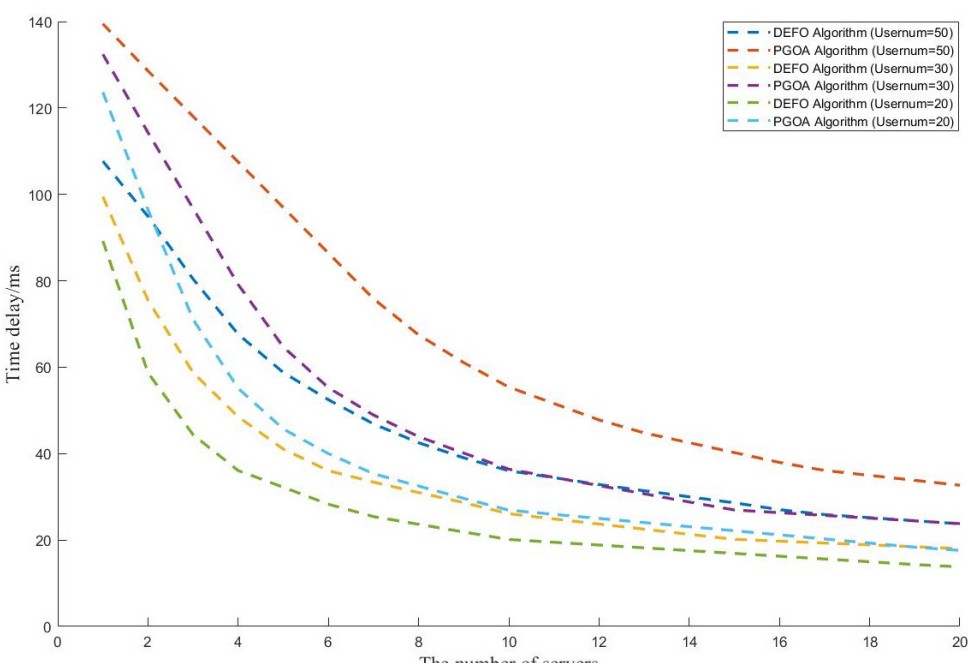

**Figure 14.** Comparison between the number of servers and the time delay of the two algorithms.

In Figure 15, we set up four MEC servers and compared the time delay between the DEFO algorithm and the PGOA algorithm at different user sizes. Compared with the PGOA algorithm, the DEFO algorithm has a slower delay change with the increase in user size. When the users reach 50, the delay increases by 60%.

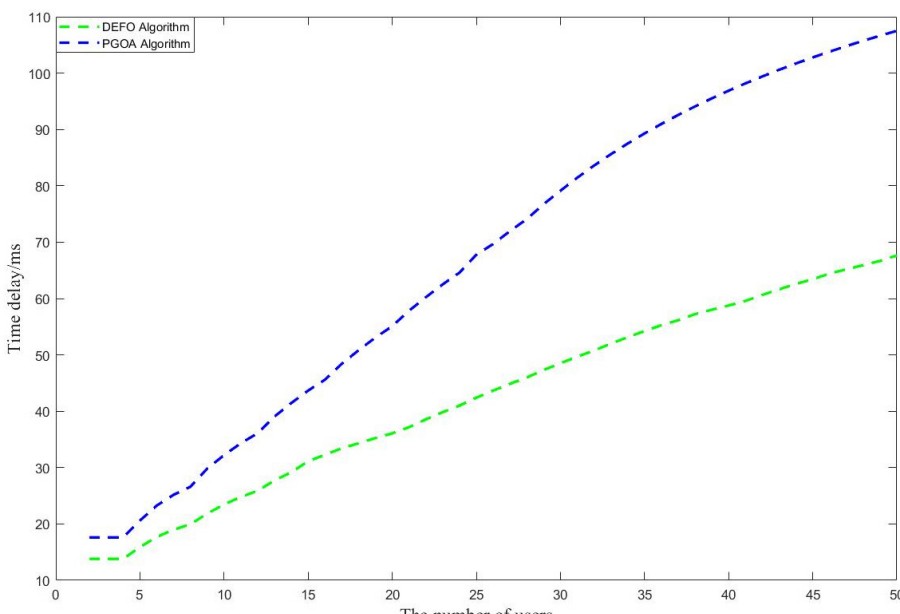

**Figure 15.** Comparison of the relationship between user size and time delay of the two algorithms.

Figure 16 task offload ratio of two algorithms with different user sizes. Whether the user unloads to the edge server depends on the revenue ratio between the overall task unloading delay and the local processing delay. From the point of view of edge operators, in the case of fixed edge network resources, the more edge users served, the greater benefits can be obtained. It can be seen from the figure that compared with the PGOA algorithm, the DEFO algorithm can improve the unloading efficiency by about 10%. However, as the number of users increases, the unloading efficiency of the DEFO algorithm also declines. At this time, we can divide more sub-tasks to improve the unloading efficiency of the system.

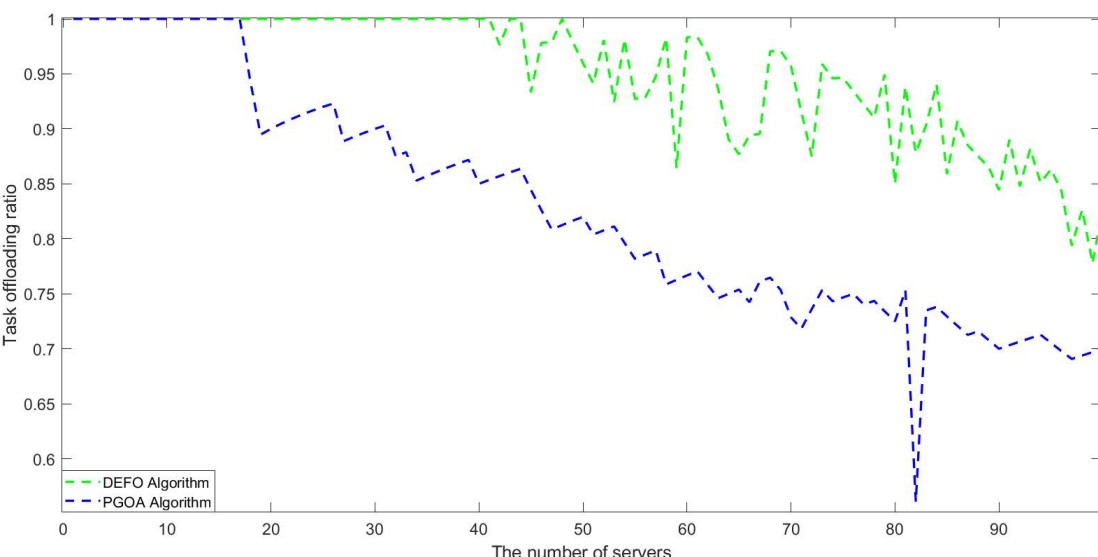

**Figure 16.** Comparison of the relationship between the number of servers and the offload rate of the two algorithms.

From the above experiments, we can draw the following conclusions:

(1) Within a certain numerical range, we increase the number of sub-tasks of the task, which can effectively improve the utilization of the network's server resources. Therefore, the DAG task model can better combine the advantages of parallel computing compared

with the chain task call graph. Taking into account the parallel characteristics of complex applications in the urban environment, mobile applications can be divided into more fine-grained task modules, and the number of sub-tasks can be increased to a greater extent, thereby effectively reducing equipment energy consumption and latency and improving offloading efficiency.

(2) In the UAV edge network, we believe that the CEFO algorithm can be applied to urban environments with small user scenarios, such as suburbs. Due to its low computing cost, the DEFO-based algorithm can be applied to places with large user scale, such as business districts and school districts.

## 5. Conclusions

In this paper, based on the UAV scene in the urban environment, we construct a fine-grained task offloading model combined with MEC, which fully considers the mobility performance of UAVs, the pre- and post-dependency of fine-grained tasks, and environmental resistance. On the basis of the model, two improved DEFO intelligent algorithms are proposed. Through experimental simulation, the proposed algorithm is compared with previous algorithms, and it is concluded that the unloading rate is improved by about 10% in numerical value, and the delay can be improved by about 60% when the user size reaches 50. We believe that DEFO algorithm is based on game theory and is more competitive in network architecture. In the construction of the computational offloading model, we constrain the energy of the user equipment and the UAV so that it does not exceed the maximum energy consumption value of the equipment, which can better meet the energy-saving and time-saving requirements of green cities.

**Author Contributions:** Conceptualization, S.Y.; methodology, S.Y. and H.Z.; software, S.Y.; formal analysis, C.M.; investigation, C.M.; resources, C.M.; data curation, S.Y. and H.Z.; writing—original draft, S.Y.; writing—review and editing, C.M.; validation, S.Y.; supervision, C.M.; project administration, C.M. All authors have read and agreed to the published version of the manuscript.

**Funding:** This research received no external funding.

**Institutional Review Board Statement:** Not applicable.

**Data Availability Statement:** Not applicable.

**Acknowledgments:** Thanks to Li Ying and Chen Yujie of the Armed Police Engineering University for their help and support and to Zhang Yiwen from the School of Information Engineering of Armed Police Engineering University for their help in technical support. The authors declare that this article has no conflict of interest with other organizations and individuals.

**Conflicts of Interest:** The authors declare no conflict of interest.

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
