# Peer review of "MEC-Enabled Fine-Grained Task Offloading for UAV Networks in Urban Environments"

_sustainability, doi:10.3390/su142113809_

Round 1

Reviewer 1 Report

La combinación de tecnología UAVs y MEC se ha convertido en una buena solución para alcanzar los objetivos de ahorro energético y baja latencia para las ciudades inteligentes. Si bien este es un artículo técnico interesante, donde se amplían las capacidades de los UAV, este artículo no está relacionado con los objetivos de la revista y no es compatible con ninguno de los temas del área de sustentabilidad.

Se recomienda ser enviado a una revista sobre ciencias de la computación, matemáticas computacionales, inteligencia artificial, sistemas de automatización y control, teoría, métodos y aplicaciones interdisciplinarios, sistemas de datos e información o aplicaciones de drones.

-----

The combination of UAV and MEC technology has become a good solution to achieve the goal of energy saving and low latency in smart cities. While an interesting technical article that expands the capabilities of UAVs, the article is not related to the objectives of the magazine and does not support any topic in the sustainability field.

Recommended submissions are to journals in computer science, computational mathematics, artificial intelligence, automation and control systems, theory, interdisciplinary methods and applications, data and information systems, or UAV applications.

Author Response

We hope the reviewers will reconsider our article and make the following statements:

  1. At present, with the increase of the number of mobile terminals in urban environment, cloud computing environment has increased the burden. Therefore, we use mobile edge computing mode combined with UAV to filter and analyze data in urban environment and make full use of idle resources in the edge environment. It can not only save energy and time, but also improve the task processing and timely response capability of edge devices. Reduce the pressure of edge devices transmitting network data to the cloud. We believe that this will contribute to the sustainable development of the city.
  2. For the proposed task unloading model, we combine the software definition network (SDN) technology and propose an improved task unloading system based on the earliest completion unloading (EFO) algorithm. At the same time, we have run the recommended unloading system in the UAV sensor. Through experiments, the algorithm can reduce system delay well under the premise of meeting energy constraints. Achieve the purpose of saving energy and time.

The innovation of this paper lies in:

  1. Combine MEC with UAV network, and use SDN technology to manage UAV network to prevent resource and energy waste caused by uneven deployment of UAV.
  2. An improved task offloading system based on the new earliest completion offload (EFO) algorithm is proposed, and the performance of the algorithm is compared through experiments on the UAV platform. Through the comparison of the results, the superiority of the algorithm in the UAV assisted MEC system is proved.

Therefore, we believe that the topic meets the requirements of the journal and hope that reviewers can consider further processing of the article.

Reviewer 2 Report

1.        This paper proposes two EFO algorithms, DEFO and CEFO. The previous simple description can be practiced in multiple scenarios. It is suggested that the following examples can be used to point out the display scenarios applicable to the two algorithms, rather than the situation description.

2.        The paper proposes to use the thinking of graph theory in mathematics to solve the problem of task partitioning, distinguish subtasks that can be assigned to remote servers, and suggest adding a detailed description of task splitting, which can be enumerated.

3.        Unmanned aerial vehicle ( UAV ) scheduling problem, it is suggested that a graph to illustrate, only through the algorithm and text description is not enough.

4.        Subtasks need to be frequently passed and offloaded in the UAV network, the article can provide offloading efficiency, a task split molecular task in the network transmission frequency and other data.

5.        Article 2.2 gives a detailed transmission power algorithm, transmission loss algorithm, UAV flight energy consumption formula, it is recommended that the UAV as an edge server can carry the energy description.

6.        Article 4.1 points out that at present, the server that UAV can carry is limited, and there is an irreconcilable contradiction between weight and processing capacity on UAV, including communication module. The two algorithms proposed in this paper are based on UAV network, so it is suggested to provide a solution to this contradiction.

7.        In the graph drawn in 4.2, the number of horizontal and vertical axes is not given in numerical units.

8.        The article points out that the user 's task split numerator task, sub-task can choose UAV server or ground server, general task processing requires UAV server and ground server cooperation, the specific way of cooperation can be pointed out in the article.

Author Response

Response:

For modification 1:In the section 4.3 of the article, we distinguish the differences between CEFO algorithm and DEFO algorithm, and introduce different scenarios where CEFO algorithm and DEFO algorithm are applicable.

For modification 2:In Section 2.1, we added examples of task division methods.

For modification 3In 2.2, we compared the priority allocation of UAVs with and without priority, which more intuitively shows the task scheduling process of UAVs.

For modification 4In the conclusion, we have improved Figure 15, explored the relationship between the number of servers and the task unloading rate, revealed the performance difference of the two algorithms in the unloading efficiency, and reflected the good performance of the algorithm proposed in this paper.

For modification 5We add the description of UAV energy in 2.2 to make the model more accurate.

For modification 6The maximum load of the UAV platform we use can reach 1.2kg, while the load of the computing platform is only 140g. Therefore, the UAV is capable of carrying the computing platform and providing services for mobile users.

For modification 7In the figures and tables in Section 4.2, we added mathematical units to the quantities that need mathematical units.

For modification 8We modified the figure in Section 2, and introduced the cooperation between UAV and ground base station. That is, when the ground base station is busy, the UAV can be used as a MEC server for calculation. When its computing capacity is insufficient to provide services, it can be pre processed and uploaded to the ground base station for processing.

Reviewer 3 Report

Title:MEC-enabled fine-grained task offloading for UAV networks 3 in urban environment

Comment1:  Abstract should include numerical results.

Comment2. in 2. UAV Edge Network Modeling, the figure 1 can be improved.

Comment3: system model block diagram can be added in 2.1. System Model

Comment 4: Figure 3. Program diagram of the EFO improve visibility

Comment 5: Conclusion  should have numerical results and contribution should be compared with previous work.

Author Response

Response:

For modification 1:We add specific numerical values in the summary and conclusion, which shows the advantages of the algorithm more intuitively.

For modification 2:We have modified UAV Edge Network Modeling in the figure 1 and added its communication mode.

For Modification 3: A text description of the block diagram of the system model in Section 2.1 was added to describe the process of model building in more detail.

For Modification 4: We describe and introduce the block diagram of the EFO algorithm.

For Modification 5: In summary, we compare the DEFF algorithm with the PGOA algorithm in terms of latency and offloading efficiency. Finally, we show that DEFO is about 60% more efficient than PGOA in terms of latency and 10% more efficient in offloading.

Reviewer 4 Report

The article is on a very good level.

I suggest to describe the problematics more in detail in paragraphs: 254-262

 The chapter title 2.2 should go to next line.

The chapter title 3.2 should go to next line.

The chapter title 4 should go to next line.

 Sharper text, axes and whole figure description including thicker line choice of the graphic in figures 9, 10, 11, 12, 13

Consider whether the position of equations, especially eq. 12, 14, 16 are not too close to the text.

 In Conclusion describe more the fulfilled goals and benefits of designed solution.

Author Response

Response:

For revision 1: We added a description of subtask structure and task division in 254-262 to enrich this paper.

For revision 2: We modified the UAV edge network modeling in Figure 1 and added its communication mode.

Revision 3: We adjusted the title and image of the article.

Round 2

Reviewer 1 Report

Reviewing the comments and evaluations of the other reviewers, they accept the article, I reviewed the objectives of the special issue to which it is being submitted and it seems that it is relatively compatible with the theme of the special issue. Although saving that the magazine works on sustainability issues, in this special edition, this topic of electronic sciences, software and computing is included, therefore, based on the joint assessment of the 3 alternate reviewers, I accept it to be published in this version.